# Learning context shapes bimanual control strategy and generalization of novel dynamics

**Jonathan Orschiedt**[1], **David W. Franklin**[1,2,3]*

**1** Neuromuscular Diagnostics, Department Health and Sport Sciences, TUM School of Medicine and Health, Technical University of Munich, Munich, Germany, **2** Munich Institute of Robotics and Machine Intelligence (MIRMI), Technical University of Munich, Munich, Germany, **3** Munich Data Science Institute (MDSI), Technical University of Munich, Munich, Germany

* david.franklin@tum.de

**Data Availability Statement:** The raw data underlying all figures is published in Figshare under the following link: https://doi.org/10.6084/m9.figshare.22895378.v2.

## Abstract

Bimanual movements are fundamental components of everyday actions, yet the underlying mechanisms coordinating adaptation of the two hands remain unclear. Although previous studies highlighted the contextual effect of kinematics of both arms on internal model formation, we do not know how the sensorimotor control system associates the learned memory with the experienced states in bimanual movements. More specifically, can, and if so, how, does the sensorimotor control system combine multiple states from different effectors to create and adapt a motor memory? Here, we tested motor memory formation in two groups with a novel paradigm requiring the encoding of the kinematics of the right hand to produce the appropriate predictive force on the left hand. While one group was provided with training movements in which this association was evident, the other group was trained on conditions in which this association was ambiguous. After adaptation, we tested the encoding of the learned motor memory by measuring the generalization to new movement combinations. While both groups adapted to the novel dynamics, the evident group showed a weighted encoding of the learned motor memory based on movements of the other (right) hand, whereas the ambiguous group exhibited mainly same (left) hand encoding in bimanual trials. Despite these differences, both groups demonstrated partial generalization to unimanual movements of the left hand. Our results show that motor memories can be encoded depending on the motion of other limbs, but that the training conditions strongly shape the encoding of the motor memory formation and determine the generalization to novel contexts.

## Author summary

Using cutlery, buttoning up a shirt, or cooking a meal requires precise coordination between two hands. These daily activities seem effortless, as they are based on well-adapted motor memories covering a wide space of experienced states. We demonstrate that the sensorimotor control system creates a motor memory of one limb using the experienced states of the other limb. Presentation of evident or ambiguous information about this relation between the two limbs shaped the bimanual control by changing the extent to which kinematic information of each arm which was used to control subsequent

**Funding:** The author(s) received no specific funding for this work.

**Competing interests:** The authors have declared that no competing interests exist.

movements. Importantly, bimanual motor memories are only partially transferred to unimanual actions, likely engaging different neural processes. This has strong implications for rehabilitation techniques that employ bimanual training.

## Introduction

The human sensorimotor control system has extraordinary abilities to flexibly perform skilled actions. Whether it is coordinating a throw in judo or simply eating with a knife and fork, humans are able to coordinate and adapt to the complex dynamics across two limbs. Despite continually changing environmental constraints, sensorimotor transformations, and dynamics, humans perform skillful actions which require a high level of control. Many studies have shown that the sensorimotor control system adapts to novel dynamics by forming a predictive motor memory (or internal model) of the task [1–3]. Moreover, we form independent motor memories specific for the coordination of the two limbs [4, 5] that depend on whether we move unimanually or bimanually [6]. Further studies on neural activation patterns confirm this distinction in internal model formation [7–9]. However, measures of transfer from bimanual to unimanual contexts, and vice versa, revealed a partial compensation [6, 10, 11] indicating an overlapping component between internal models for these two contexts. Evidence from unimanual studies underlines the idea of a task-specific, local internal model formation. Adaptation to novel dynamics forms state-dependent representations: states experienced during learning are used to update the internal model and allow for adaptation to the novel environment [1, 12–15]. This has been powerfully underlined by the study of Gonzales-Castro, Monsen and Smith [16], who showed that the sensorimotor control system relies on motion-referenced learning. That is, internal model updates between trials were most efficient for trials where the plan for the current trial matched the trajectory from the previous trial. However, learning does not only occur for the trained movement [3], but also generalizes to movements across similar states [1, 17–21], where the amount of generalization decreases as the states diverge further from the trained conditions [11, 16, 22–25]. These generalization results have suggested that learning occurs through the tuning of Gaussian-like units or neural basis functions [26–29], which can be combined to allow for flexible actions in new environments.

The number of possible factors and states influencing the encoding and adaptation of motor memories multiplies in bimanual tasks. Bays and Wolpert [30] showed that the feedforward force generation of a stationary arm adapting to a force perturbation could be tailored to the movement direction of the opposite arm. Using a similar bimanual paradigm, Jackson and Miall [31] investigated the adaptation to a force perturbation, which was proportional to the movement speed of the contralateral arm performing a reaching movement. Their participants reduced the final end-point error quickly and showed an appropriate scaling of the predictive force to changes in movement velocity. They demonstrated that predictive forces by one arm could be generated based on the state-dependent encoding of the speed of the movement of the other arm. However, Jackson and Miall [31] did not assess how the contralateral predictive response would be shaped if both arms would do a reaching movement, where the states experienced by both arms could influence the formation of the motor memory. Yokoi et al. [11] demonstrated that directional changes in movements of both arms led to gaussian-like decay in predictive force, supporting the hypothesis of overlapping neural control processes [6] and the flexible encoding of kinematics of both arms to perform bimanual actions. Taken together, the sensorimotor control system is able to produce highly flexible motor commands simultaneously for our arms, based on the encoding of either one or both arms. However, it remains

unclear how the sensorimotor control system associates the learned memory with the experienced states in bimanual movements. More specifically, how and to what extent does the sensorimotor control system combine the states of the left and right arms in the encoding of bimanual motor memory?

Here, we tested the motor memory formation with a novel paradigm which required the encoding of the kinematics in velocity space of the opposite, right hand to produce an adequate, predictive force on the left hand. While one group (the *ambiguous* group) was trained on matching movements with similar speeds and lengths for both arms, the other group (the *evident* group) had non-matching movements with different speeds and lengths for the left and right arms. Hence, the latter (evident) group faced different states between arms within each trial which allowed them to experience the evident relationship between the right-hand kinematics and the force field on the left hand. In contrast, the matching movement speeds of the ambiguous group provided little information about the appropriate encoding as the states of both arms were similar. Differences between arms could only be sensed through variability in the speeds of the two hands, hence the sensory information provided was ambiguous. To assess the underlying representation, specifically the amount of right-hand and left-hand encoding, we tested how participants generalized to novel combinations of the trained conditions (speeds). Furthermore, we removed the state information of the right hand in some trials to observe how the formed motor memory would cope with the lack of information and transfer to unimanual trials. Overall, we were able to assess how the sensorimotor system combines the experienced states from both arms to produce an appropriate motor command and discovered that this bimanual encoding is formed during exposure to a specific environment.

## Results

The aim of the current study was to understand how the sensorimotor control system associates the learned motor memory with the experienced states in bimanual movements. We trained two groups in different conditions, in both of which they had to adapt their movements to a velocity-dependent force field imposed on their left hand, where the forces experienced depended on the movement of their right hand (Fig 1). The evident group trained on conditions in which the participants would clearly experience the left-hand forces depending on the right-hand velocity (conditions: $b_{fs}$, $b_{mm}$ and $b_{sf}$). On the other hand, the ambiguous group trained on conditions in which this relationship between the left-hand forces and right-hand movements was not clearly apparent (conditions: $b_{ff}$, $b_{mm}$ and $b_{ss}$), but could only be sensed through variability in the speeds of the two hands. After adaptation to the force fields in these conditions, we tested the representation of the motor memory by measuring the participants' predictive forces in novel conditions not experienced during the exposure phase (Fig 1D). We hypothesized that the evident group would extract the relationship between the left-hand forces and right-hand kinematics, and therefore should generalize the motor memory to the novel conditions (development of predictive force based on right-hand kinematics). In contrast, we hypothesized that the ambiguous group will be unable to extract this relationship, and therefore express a generalization pattern relying on the kinematics of the left hand or a combination of both and hence produce an inappropriate motor output.

To assess the amount of information participants could potentially extract between both hands, we compared the differences in velocity profiles between right and left hand (Fig 2) and conducted a repeated measures ANOVA for each group separately. For both the evident group ($F_{5,35} = 190.407$, $p < 0.001$, $\eta_p^2 = 0.965$) and the ambiguous group ($F_{1.749,12.240} = 173.675$, $p < 0.001$, $\eta_p^2 = 0.961$, Greenhouse-Geisser corrected), the conditions showed a significant main effect. The conditions $b_{sf}$ and $b_{fs}$ for the evident group showed a clear difference during

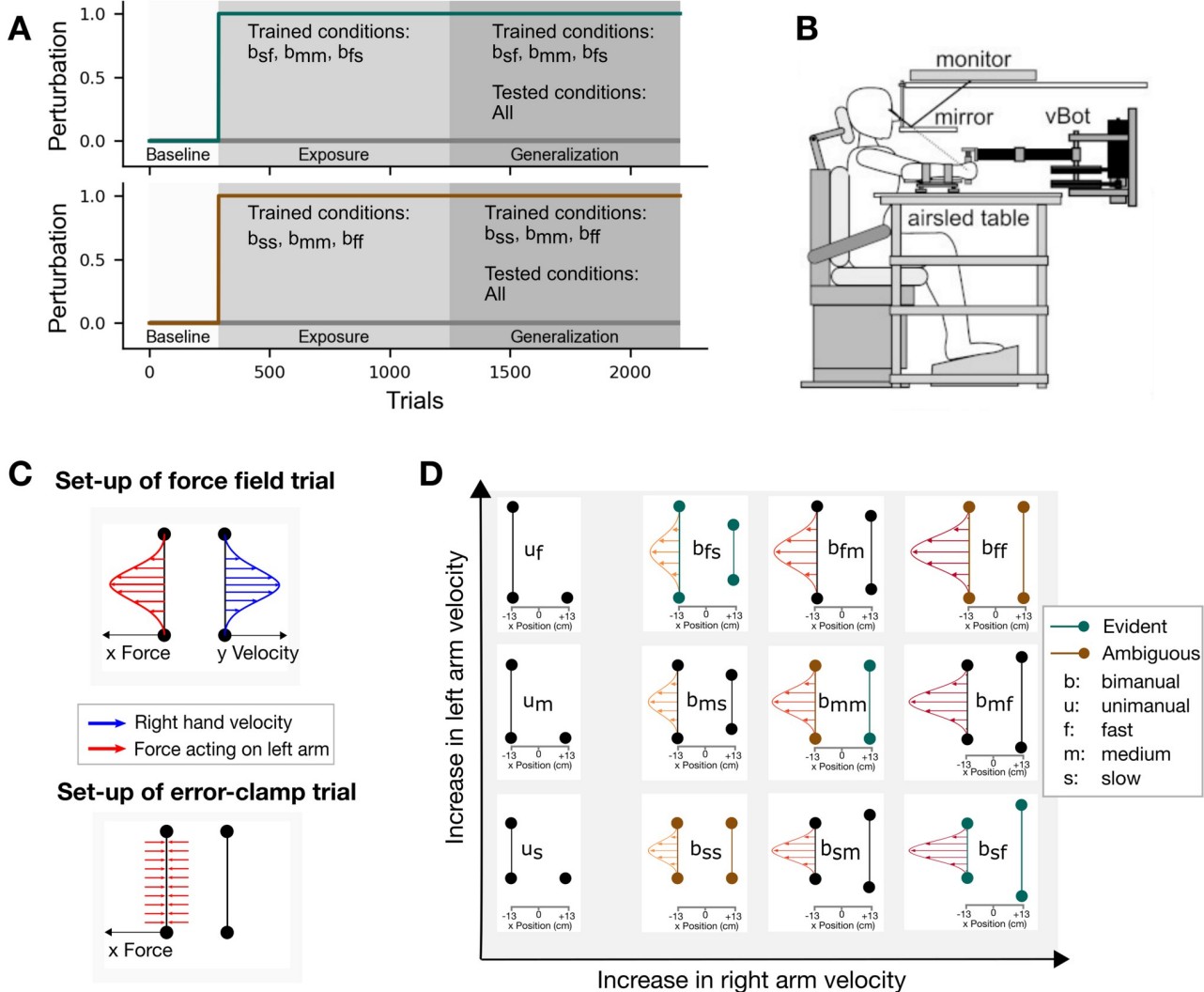

**Fig 1. Experimental set-up for both groups.** (**A**) The experiment was divided into three phases: Baseline, Exposure and Generalization. Each block consisted of 48 trials and overall 2208 trials were completed. During the exposure phase, participants were confronted with the three depicted training conditions as well as error clamp trials to assess adaptation. In the generalization phase, the same three training conditions were still used, however, the interspersed error-clamp trials were applied to all conditions, both unimanual and bimanual. (**B**) The vBOT was used to apply state-dependent forces together with a virtual environment presented via the monitor. (**C**) The force field, which was applied to the left arm, was determined by the velocity of the right arm during bimanual movements. Error-clamp trials were mechanical force channels, which prohibited any lateral movement of the left hand and allowed us to measure lateral forces. (**D**) Nine bimanual and three unimanual conditions were used to assess generalization. The colored, green and brown conditions represent the trained conditions for the evident and ambiguous group, respectively. The curved lines depict the required amount of force in case of left-hand encoding during the generalization phase.

the exposure phase (Left $b_{sf}$ vs. Right $b_{sf}$: $p < .001$; Left $b_{fs}$ vs. Right $b_{fs}$: $p < .001$), whereas all three conditions in the ambiguous group showed only small differences between hands, which were similar to the condition $b_{mm}$ for the evident group all $p > 0.05$). The small differences can be explained by different temporal patterns between the right (dominant) and the left hand. Interestingly, for all conditions with matched movements of the left and right hands, the right hand slightly preceded the movement of the left hand across all our participants. Moreover, both groups complied with the desired peak velocities of the experimental design (Fig 2G).

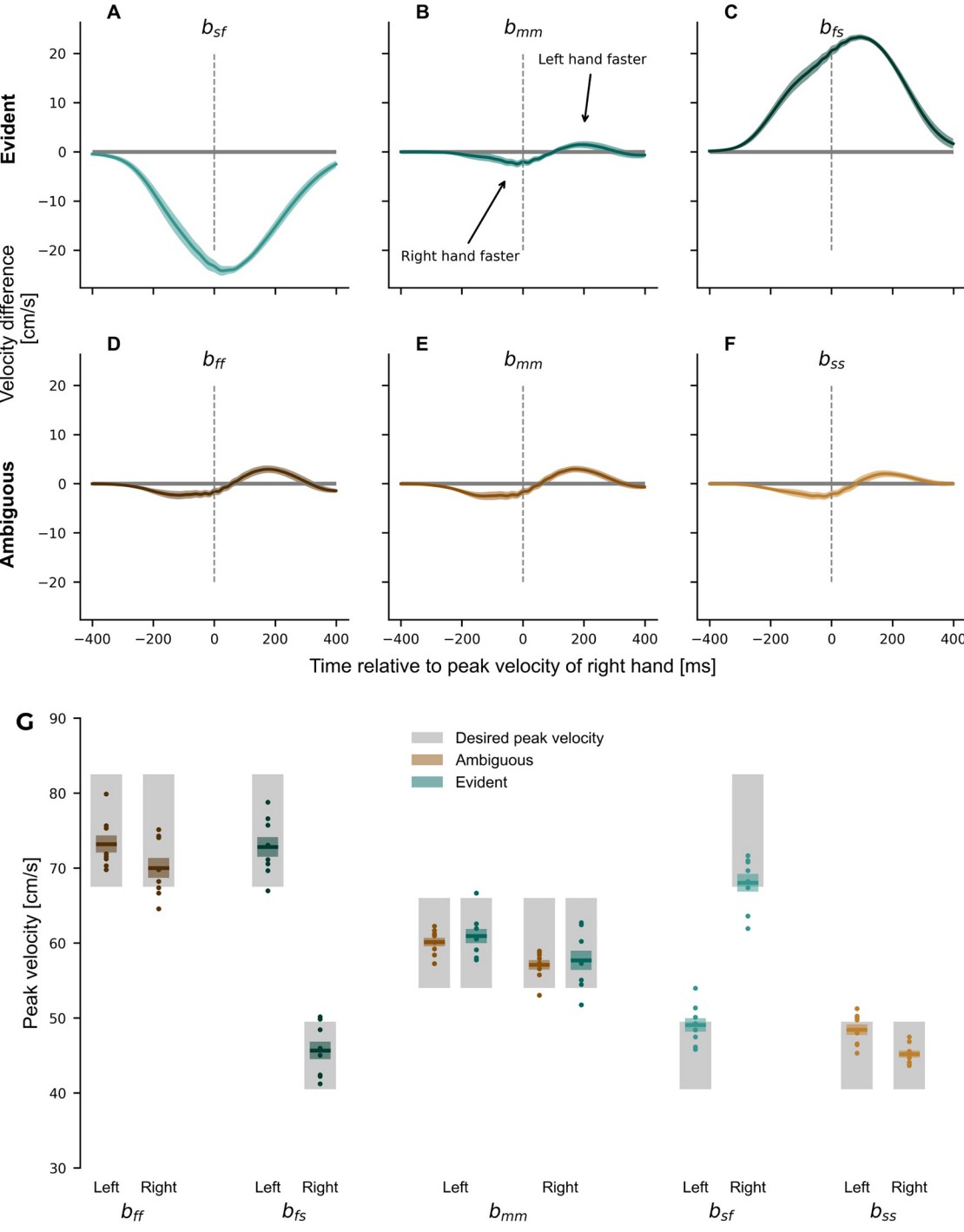

**Fig 2. Hand velocities during the exposure phase.** (**A-F**) Difference in velocity profiles between left hand and right hand for all trials during the exposure phase. (**A-C**) The trained conditions for the evident group. (**D-F**) The trained conditions for the ambiguous group. The force profiles depict the mean difference (left-hand velocity—right-hand velocity) between hands across subjects. While the ambiguous group showed only minor differences between hands, the evident group experienced major differences in conditions $b_{sf}$ and $b_{fs}$. (**G**) Compliance of peak velocity with experimental design. Mean ± standard error of the mean (colored region) of the right and left-hand peak velocities for each of the training conditions. Dots represent individual participants. The grey-shaded region indicates peak velocities that were considered successful for each condition. Participants adhered to the respective desired peak velocities from the experimental setup.

Consequently, only little information on the relationship between right-hand velocity and the perturbing force could be extracted for the ambiguous group during exposure.

## Adaptation to novel dynamics

In the baseline phase, participants made straight movements to the targets as no lateral forces were applied in the null field. When the force fields were applied in the exposure phase, this produced large lateral errors which were gradually reduced. Finally, in the generalization phase the representation of the motor memory was probed by testing generalization to untrained conditions. To verify that both groups were able to learn the imposed dynamics, we compared the baseline movements in the null field with the late movements in the exposure and generalization phases (Fig 3). We used four different parameters, namely the maximum perpendicular error, the force compensation, and the peak and shape of the force profiles, to assess motor learning in response to the novel dynamics.

In terms of kinematics, the maximum perpendicular error showed a sharp increase in the initial exposure to the force field (Fig 3A), which was gradually reduced with continued exposure to the force field. These later straighter movements towards the target indicated adaptation to the force fields. Both groups adapted fairly quickly to the novel dynamics, reaching steady-state levels of kinematic error within the first half of the exposure phase, with maximum perpendicular error levels remaining around this level for the rest of the experiment. We performed a repeated measures ANOVA with within-subjects factor *phase* (early exposure, late exposure and late generalization) and between-subjects factor *group* (Fig 3B). There was a significant effect for phase ($F_{1.229,17.201} = 35.023$, $p < 0.001$, $\eta_p^2 = 0.714$), but no effect between groups ($F_{1,14} = 0.079$, $p = 0.783$, $\eta_p^2 = 0.006$) and no interaction ($F_{1.229,17.201} = 0.100$, $p = 0.806$, $\eta_p^2 = 0.007$). Post-hoc Bonferroni pairwise comparison indicated a decrease from early exposure to late exposure by 2.18 cm ($p < 0.001$), but no further decrease between late exposure and late generalization (Mean difference of 0.042 cm, $p = 0.777$).

To quantify the amount of predictive adaptation to the novel dynamics, we examined the force compensation on random error clamp trials. In the baseline phase, force compensation was close to zero with 7.62±2.27% and 6.88±2.77% for the evident and ambiguous group, respectively. Both groups showed similar responses to the introduction of the force field with a sharp, fast increase in force compensation during the initial exposure block (Fig 3C and 3D). In subsequent blocks, the performance increase slowed down and the learning curve was asymptotic at 79.52±3.83% for the evident group and 86.88±2.85% for the ambiguous group in the late exposure phase. A repeated measures ANOVA revealed a significant main effect for this change across the phase ($F_{2,18} = 125.963$, $p < 0.001$, $\eta_p^2 = 0.933$), but not for group ($F_{1,9} = 0.003$, $p = 0.960$, $\eta_p^2 = 0.003$) or interaction ($F_{1,14} = 2.902$, $p = 0.111$, $\eta_p^2 = .041$). Post-hoc Bonferroni tests showed that the average increase of 76.92% between baseline and exposure was significant ($p < 0.001$), but the slight decrease by 8.68% between late exposure and late generalization was not significant ($p = 0.120$).

We further quantified this adaptation by examining the development of the force profiles during exposure to the force fields (Fig 4). In the baseline phase (Fig 4A and 4F), force traces showed only minor deviations from zero, which can be explained by slightly curved movements, which produced small lateral forces. During the first block of exposure, there was an increase in amplitude compared to baseline performance (Fig 4B and 4G). With continued exposure, the force amplitudes further increased and diverged for the three different trained conditions. By the end of the generalization phase we see a clear modulation of the predictive forces according to the conditions for both groups (Fig 4D and 4I) that is a scaled version of

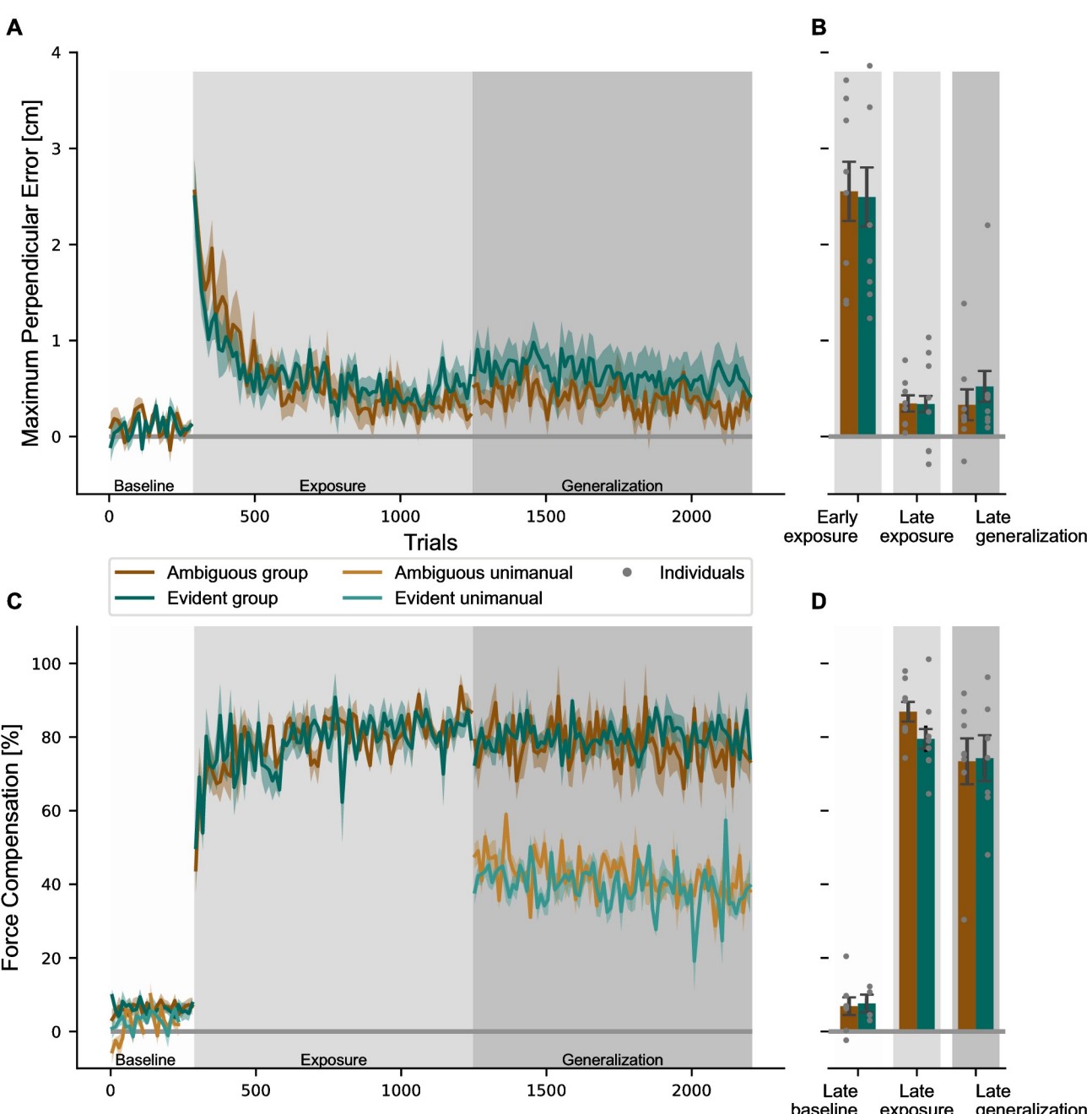

**Fig 3. Development of the motor learning parameters *maximum perpendicular error* and *force compensation* during the experiment.** (**A**) The maximum perpendicular error throughout the experiment is shown for the three training conditions for the ambiguous (brown) and the evident (green) groups. After the baseline phase in the null field, the maximum perpendicular error increased initially with the introduction of the perturbation in the exposure phase but rapidly decreased with continued exposure to force fields, remaining at a constant level throughout the generalization phase. Values are shown in trial bins of 12 and reported as mean values across participants (solid line) ± standard error of the mean (shaded region). (**B**) The mean maximum perpendicular error is shown for the first bin of 12 trials in the exposure phase, the last bin in the exposure phase and the last bin in the generalization phase (error bars represent standard error). Individuals are represented by grey dots. (**C**) The force compensation is shown across the experiment for the bimanual trials (dark brown and dark green) and unimanual trials (light brown and light green), which were only presented in baseline and generalization phases. Bimanual force compensation increased sharply after introduction of the force field and stayed high throughout the exposure and generalization phases. Note, that in the generalization phases, the bimanual values include all of the bimanual conditions, not only the trained ones. Unimanual force compensation increased from baseline to generalization phase, but did not reach bimanual adaptation levels. (**D**) Force compensation for last bins in baseline, exposure and generalization phases.

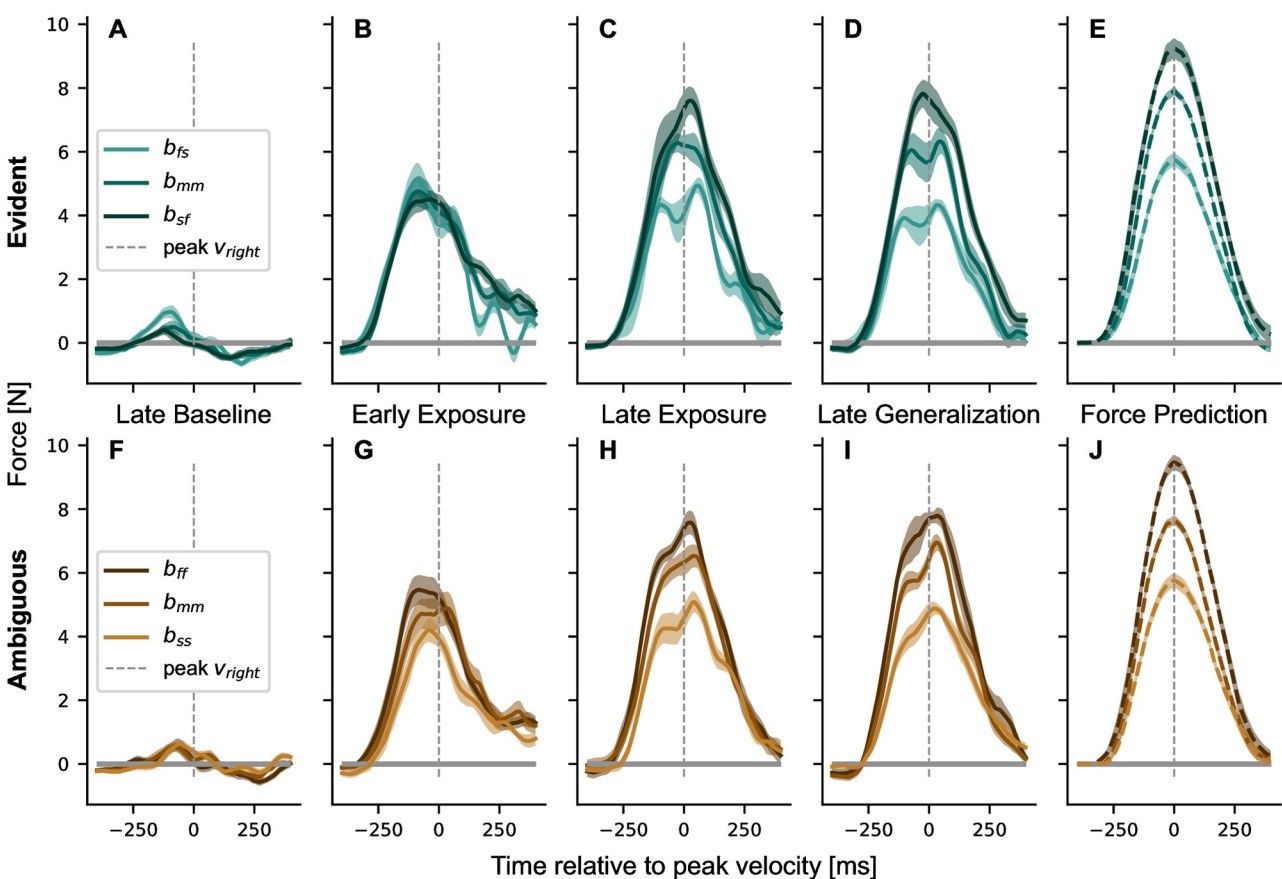

**Fig 4. Changes in lateral force profile of the left hand during the experiment for the trained conditions.** (**A** to **E**) The evident group. (**F** to **J**) The ambiguous group. The last column (**E**, **J**) represents the required forces for complete adaptation. The force profiles represent mean values across participants and blocks of 48 trials and were aligned to the peak speed of the right hand. While the lateral force was generic across conditions in early adaptation, longer exposure led to clear differentiation between conditions according to the required force. Note that the slow movement of the left hand in the evident group ($b_{sf}$) required high forces, while it required low forces in the ambiguous group ($b_{ss}$) (e.g., in **D** and **I**).

the perfect compensation for each of the conditions (Fig 4E and 4J). To evaluate the tuning between conditions, we performed a two-factor ANOVA with repeated measures (factors *condition* and *stage*) for each experiment, which revealed a significant effect for both factors in both the evident (factor *Condition* $F_{2,14} = 33.617$, $p < 0.001$, $\eta_p^2 = 0.828$; factor *Stage* $F_{3,21} = 69.889$, $p < 0.001$, $\eta_p^2 = 0.909$) and ambiguous (factor *Condition* $F_{2,14} = 47.413$, $p < 0.001$, $\eta_p^2 = 0.871$, *Stage* $F_{3,21} = 109.462$, $p < 0.001$, $\eta_p^2 = 0.940$) groups. Interestingly, the forces for different conditions were not tuned to the respective force field initially during early exposure, which is underlined by post-hoc Bonferroni pairwise comparison in early exposure for both the evident ($b_{fs}$ vs. $b_{sf}$: $p = 1.000$, $b_{fs}$ vs. $b_{mm}$: $p = 1.000$, $b_{mm}$ vs. $b_{sf}$: $p = 1.000$) and ambiguous ($b_{ff}$ vs. $b_{ss}$: $p = 0.116$, $b_{ff}$ vs. $b_{mm}$: $p = 1.000$, $b_{mm}$ vs. $b_{ss}$: $p = 0.758$) groups. Following extensive exposure to the force field, the peak force values diverged, which is shown by the differences in late exposure between the trained conditions in both the evident ($b_{fs}$ vs. $b_{mm}$: $p < 0.001$, $b_{fs}$ vs. $b_{sf}$: $p < 0.001$, no difference for $b_{mm}$ vs. $b_{sf}$: $p = 0.332$) and ambiguous ($b_{ff}$ vs. $b_{ss}$: $p < 0.001$, $b_{mm}$ vs. $b_{ss}$: $p = 0.005$, no difference for $b_{ff}$ vs. $b_{mm}$: $p = 0.357$) groups.

Taken together, these results highlight the ability of individuals to adapt to different force requirements simultaneously and finely tune the lateral force to the desired force profile. Both

evident and ambiguous groups adapted to the respective trained conditions and were able to scale the lateral force output to different peak velocities. While the ambiguous group learned higher forces for faster velocities of their left hand, the evident group learned to produce higher forces for slower movements of their left hand. As the right-hand velocity determined the forces on the left hand, the evident group had to tune the force of the left hand by adjusting it to the right-hand velocity for each condition.

## Fitting of encoding models in trained conditions

Although the two experiments differed in the trained conditions, the underlying relationship of the right arm velocity influencing the perturbation on the left arm was never explicitly presented to the participants. Therefore, we hypothesized that participants could use four different encoding strategies. If participants used a right-hand encoding, participants would purely use the velocity of the right arm to produce the motor command of the left arm. Inversely, a left-hand encoding would only use left-hand velocity. As participants moved both arms, we also tested the average encoding (average between both velocity profiles) and weighted encoding (weighted average between left and right velocity profiles). We fitted the four models to the produced force profiles during the generalization phase, but only to the trained conditions and retrieved the parameters for $\alpha$ and $\omega$ (see Predicted outcomes and encoding weights) and conducted repeated-measures ANOVA for the two parameters. For $\alpha$, which represents the scaling of forces to the imposed force field, there was no significant difference between groups ($F_{1,14} = 1.512$, $p = 0.239$, $\eta_p^2 = 0.097$), but effects between models ($F_{1.359,19.021} = 98.320$, $p < .001$, $\eta_p^2 = 0.875$, Greenhouse-Geisser corrected) and an interaction ($F_{1.359,19.021} = 47.210$, $p < .001$, $\eta_p^2 = .771$, Greenhouse-Geisser corrected). Post-hoc Bonferroni tests reveal a lower $\alpha$ for the left encoding ($p < .001$ in all three comparisons), while the interaction reveals, that this was mainly driven by the evident group, which had a significantly lower $\alpha$ for left encoding compared to the other models (all $p < .001$). The results of the fitting for the weighted model reveal a significant difference between groups for the weighting parameter $\omega$ (independent t-test, $t(14) = -11.420$, $p < .001$, Cohen's $d = -5.710$). Consequently, we would assume that the evident group mostly uses the right arm velocity ($\omega = 0.935$), while the ambiguous group uses kinematic information of both arms with a tendency towards the left arm ($\omega = 0.374$) (see also Section Transfer to unimanual conditions). To assess model performance and account for differences in parameters, we calculated the Bayesian Information Criterion (BIC) for all models and compared the performance to a non-parametric model with a fixed $\alpha = 0.822$, which represents the mean $\alpha$ across all models and participants. The BIC improvements for each participant and group average are depicted in Fig 5 and show that the right and weighted encoding outperform the non-parametric, left and average encoding in the evident group ($\Delta BIC > 10$ [32]). Similarly, the weighted encoding has the lowest BIC value for the ambiguous group, followed by the average, left and right encoding (all differences $\Delta BIC > 10$). The explained variance of $R^2$ is high for all five models. Taken together, we could show that the weighted encoding model fits best to the respective trained conditions while showing a significant difference in the importance by which the groups weigh the kinematic information from each arm. In order to examine if the learned movement pattern only applied to the trained conditions, representative of a local learning function, or if participants were able to extract the relationship between forces and right-hand movements, representative of a generalized learning function, we tested the participants with novel conditions in the generalization phase.

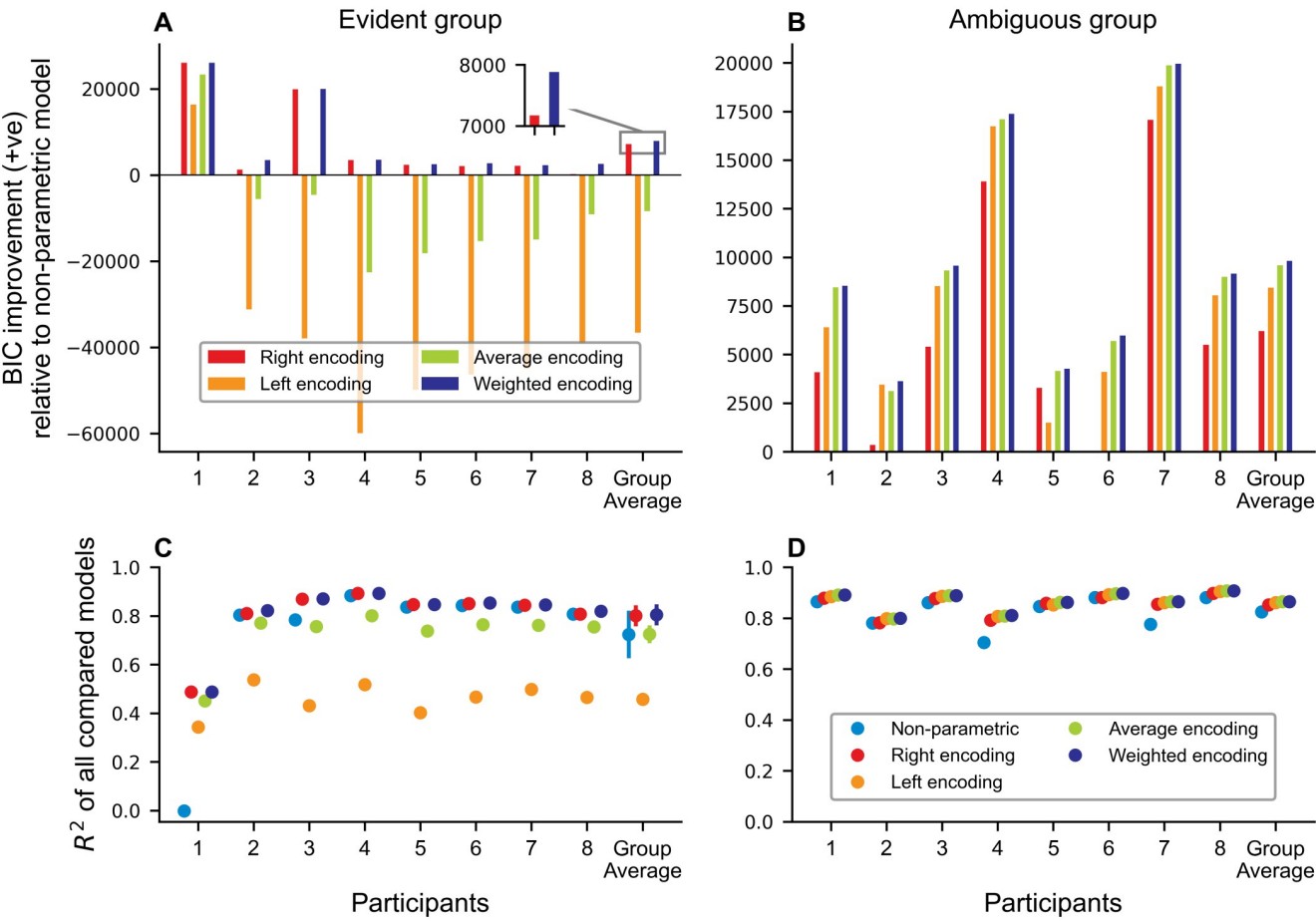

**Fig 5.** BIC model comparison for trained conditions for the evident (**A**) and ambiguous group (**B**). Values are shown as positive BIC improvements compared to a non-parametric model. The last block shows the group average for each group and the inset allows for better readability between the right and encoding model. **C** and **D** show the respective $R^2$ for each participant as well as the group average. Stronger deviations in $R^2$ in the evident group can be explained by the stark differences in the velocities between arms during the trained conditions, while there was a high correlation between movement velocities in the trained conditions for the ambiguous group. For all points the standard error of the mean is shown with a line. For most conditions this s.e. m. is generally not visible beyond the dot as the values are small.

## Generalization

The generalization phase consisted of force field trials in the respective training conditions and interleaved error clamp trials covering all conditions displayed in Fig 1D. Despite the increased number of error clamp trials in the generalization phase, there was little or no change in the overall force compensation or maximum perpendicular error (Fig 3). However, individual conditions showed strong differences in the generated peak forces (S1 Fig) and generated force compensation (Fig 6). Fig 6A and 6B outline the generated force compensation for each condition and group. We conducted a rmANOVA with "condition" as the within-subject factor and "group" as the between-subject factor. While there was no difference between experimental groups ($F_{1,14} = 0.327$, $p = 0.577$, $\eta_p^2 = 0.023$), the analysis revealed differences between conditions ($F_{3.333,46.660} = 50.928$, $p < 0.001$, $\eta_p^2 = 0.784$, Greenhouse-Geisser corrected) and an interaction ($F_{3.333,46.660} = 59.328$, $p < 0.001$, $\eta_p^2 = 0.809$, Greenhouse-Geisser corrected). Neither of the groups had a difference in force compensation between trained conditions (all

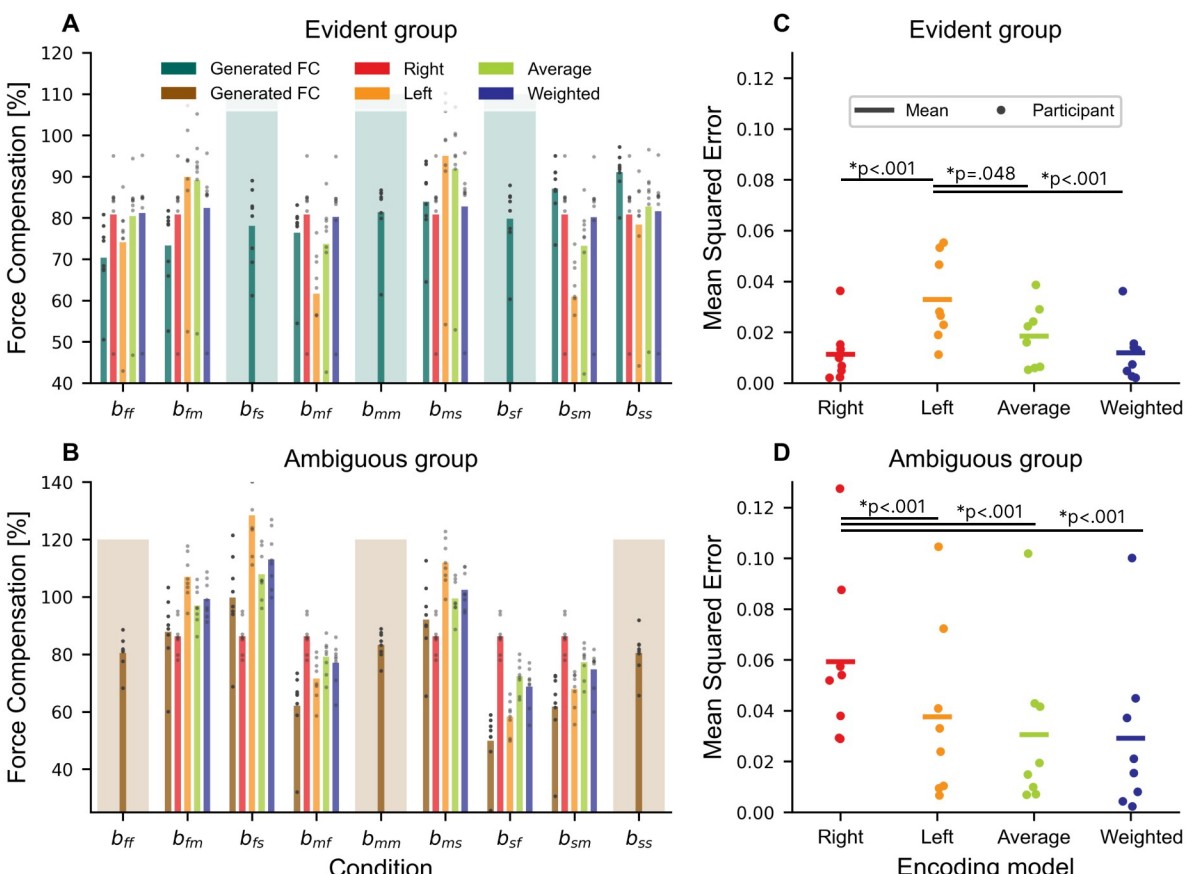

**Fig 6. Different encoding predictions for the force compensation and the quality of the estimation for bimanual generalization conditions for the evident and ambiguous group.** (**A**) Evident group. (**B**) Ambiguous group. Predicted force compensation for each encoding hypothesis and the generated force compensation for all bimanual conditions in the experiment. The predictions for left-hand encoding in the ambiguous group, right-hand encoding for the evident group and average and weighted encoding in both groups resembled best the generated force compensation in the the generalized, bimanual conditions. The mean squared error (MSE) for each encoding across all generalization bimanual conditions in the evident group (**C**) and the ambiguous group (**D**). Note that the MSE for the trained conditions are not included in these values. The weighted and average encoding fits well across both groups.

$p = 1.000$), which confirms that participants learned to tune the forces to individual conditions (compare shaded conditions in Fig 6A and 6B). However, the evident group showed significantly lower values for $b_{ff}$ (to $b_{mm}$, $b_{ms}$, $b_{sf}$, $b_{sm}$ and $b_{ss}$, all $p < .05$) and $b_{fm}$ (to $b_{ms}$, $b_{sm}$ and $b_{ss}$, all $p < .05$) as well as higher compensation for $b_{sm}$ (to $b_{fs}$, $b_{mf}$, $b_{mm}$, all $p < .05$) and $b_{ss}$ (to $b_{fs}$, $b_{mf}$, $b_{mm}$ and $b_{sf}$, all $p < .05$). This shows that the participants from the evident group did not perfectly generalize to the novel conditions. Importantly, a higher compensation does not indicate a better adaptation—when we assume a learned force compensation yielding approx. 80% in the trained conditions, the perfect generalization is indicated by the right prediction as a model of the actual underlying environment. Therefore, we would expect participants to generate a force compensation close to the red bars in Fig 6. This deviation from trained conditions and the underlying true relation of the force field is stronger for the ambiguous group, we showed bigger differences in the posthoc-Bonferroni comparison. Participants produced a force compensation, which was significantly higher for $b_{fs}$ and lower for $b_{mf}$, $b_{sf}$ and $b_{sm}$ (all $p < .001$). Consequently, the lower generalization in this group underlines, that the ambiguous group did not learn the true relation of the force field, as they were only experiencing

correlated movement velocities during training. To evaluate which encoding model explains best the generated forces, we added the predictions for each model to Fig 6. The right encoding embodies the true relation of the force field. For the evident group, the weighted and right model are similar, as the mean $\omega$ retrieved from the fit was 0.935±0.021, therefore the sensorimotor system in the evident group mostly relied on right-hand information. In contrast, predictions for the average and left encoding model showed higher deviations with increased differences between right and left-hand velocity in conditions $b_{fm}$, $b_{mf}$, $b_{ms}$ and $b_{sm}$. This trend is quantified by the mean squared error (MSE), which is lowest for right-hand and weighted encoding, followed by the average encoding in the evident group. We conducted a repeated measures ANOVA with model and group as factors for the MSE, which revealed a significant difference between models ($F_{1.342,18.788} = 12.339$, $p < 0.001$, $\eta_p^2 = 0.468$, Greenhouse-Geisser corrected), and interaction ($F_{1.342,18.788} = 19.616$, $p < 0.001$, $\eta_p^2 = 0.584$, Greenhouse-Geisser corrected), but no difference between groups ($F_{1,14} = 2.907$, $p < 0.110$, $\eta_p^2 = 0.172$). Bonferroni's post-hoc comparison revealed that effects are only significant between left-hand encoding and the other proposed models for the evident group (Fig 6C, all $p < .048$). Similarly, the weighted and average encoding in the ambiguous group is low, but the MSE for right-hand encoding was significantly higher compared to the other models (all $p < .001$). We obtained similar results for the MSE when quantifying the error between predicted and generated forces (S2 Fig). Taken together, the average and weighted encoding models fitted best across experiments. As expected and in contrast to the ambiguous group, the evident group showed a higher adherence to the right encoding. Consequently, they were at least partially able to retrieve this underlying relationship.

To further improve the evidence for the encoding models, we added a second BIC model comparison, taking all bimanual conditions into account (see Fig 7). In the evident group, the average and left-hand encoding perform worse than the non-parametric model. Previous results (compare Fig 5) showing that the weighted encoding outperforms the right-hand encoding model, are confirmed ($\Delta BIC > 10$)(Fig 7A). Moreover, the weighted encoding in the ambiguous group showed a better BIC improvement compared to left-hand, average and right-hand encoding (Fig 7B). Interestingly, these effects were observed across all participants. All models show an average $R^2 > 0.585$, hence we assume a good approximation of our models to the data. Given the presented evidence, we assume that both groups employed a weighted encoding scheme tailored to their experience during the trained conditions. The evident group adheres to an encoding scheme which strongly relies on velocity information from the right arm, which shows that they were able to at least partially retrieve the relation of the force field. In contrast, the ambiguous group combined the sensory information from both arms with a stronger influence of the left arm. These results underline that the sensorimotor system is able to variably use sensory information of both arms to produce motor commands.

## Transfer to unimanual conditions

To examine the transfer of learning from bimanual to unimanual movements, we introduced three unimanual conditions (Fig 1D) with left-hand movements while the right hand was stationary. Previous research has shown partial transfer from bimanual to unimanual conditions when the force field magnitudes depend on the velocity of the same hand experiencing the forces. However, in our task, the forces depend on the movement of the other hand (right hand). Therefore, if participants were able to learn that the force fields depended entirely on the right-hand movement velocity (right-hand encoding), then they should produce no force on the unimanual movements. Inversely, a left-hand encoding should produce forces matching the forces from bimanual trials (See red and orange lines, respectively, in Fig 8B). However,

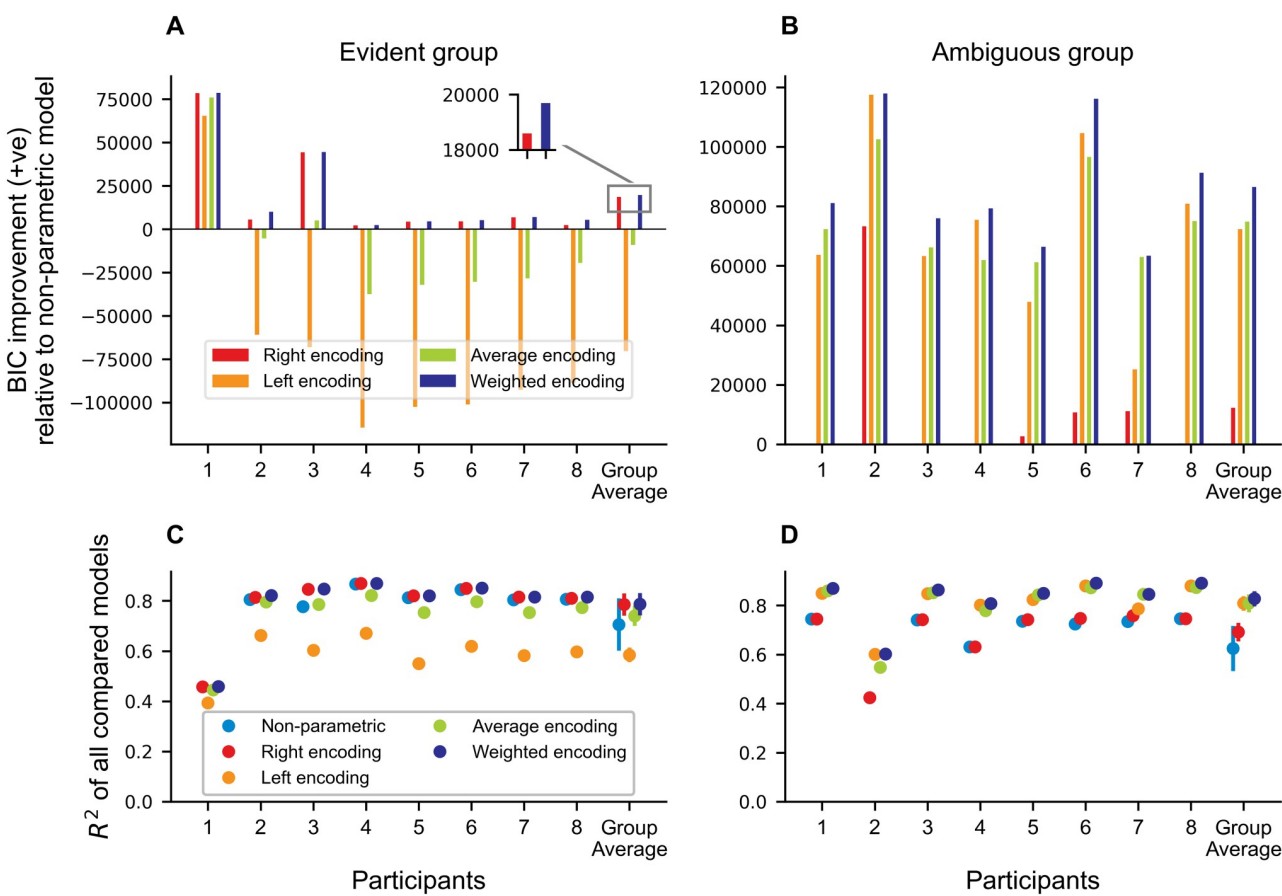

**Fig 7. BIC model comparison for all bimanual conditions as positive BIC improvement from non-parametric model. A** and **B** show the BIC for each participant and the group average for the evident and ambiguous group, respectively. The inset compares right hand and weighted encoding for better readability of the difference. The weighted encoding model has the best BIC improvement compared to the non-parametric model across both groups. **C** and **D** depict $R^2$ for each participant and group. For all points the standard error of the mean is shown with a line. For some conditions this s. e.m. is not visible beyond the dot as the values are small.

the participants of both groups produced peak forces in the unimanual conditions which were lower than during bimanual trials (Compare forces to Fig 4), and higher than a right encoding would predict. While for the ambiguous group both the average and the weighted encoding match well (Fig 9H to 9J), only average encoding is close to the generated forces in the evident group. The similarity in the fit between average and weighted encoding for the ambiguous group is explained by the mean encoding weights in the weighted condition (see Fig 8). While a fixed weight of 0.5 represents the average encoding, the weights were not fixed in the weighted encoding. Here, we fitted the weights across all trained conditions after adaptation (see Methods, Eq 6). The ambiguous group showed a mean weight of $0.374 \pm 0.045$, which was significantly different from the evident group with $0.935 \pm 0.021$ (independent t-test, $t(14) = -11.420$, $p < 0.001$, Cohen's $d = -4.410$). Hence, the ambiguous group relied more on the kinematic encoding of both arms with a slight tendency towards the left to predict the next trial, while the evident group relied strongly on the right arm with only a small contribution of the left arm in bimanual trials. Confronted with the loss of information from the right arm in unimanual trials, the evident group seems to employ the average encoding. When the average encoding depicts the standard, naive state in bimanual actions, we should also detect a gradual

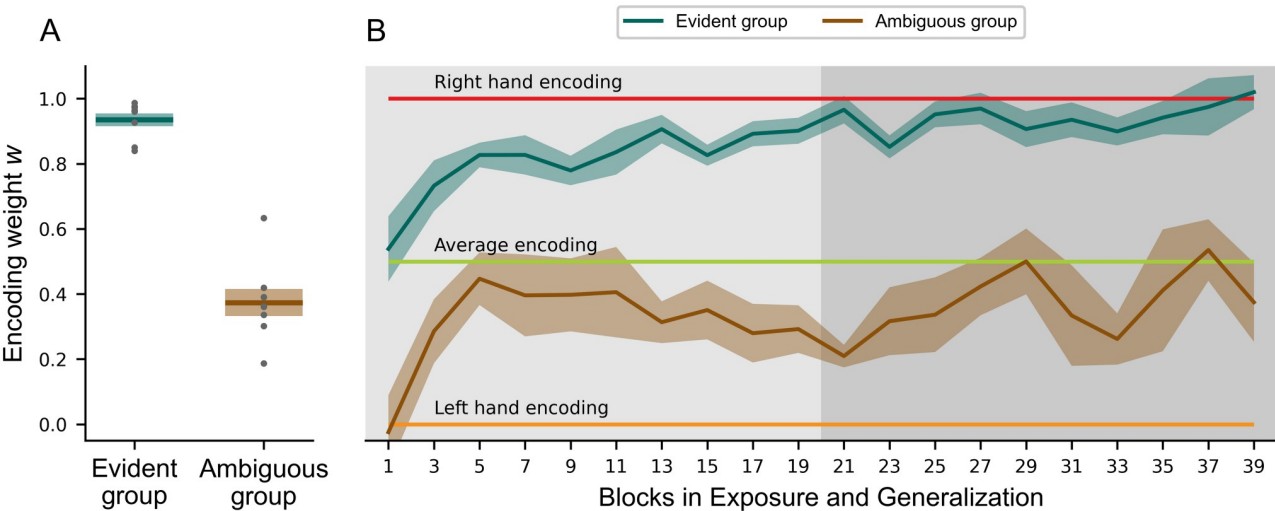

**Fig 8. Differences in encoding weights between the ambiguous and evident groups.** (**A**) Mean encoding weights in generalization phase. (**B**) The development of weighted encoding weights across blocks (96 trials/block) during exposure (light grey) and generalization phase (dark grey). The dark green and brown lines represent the mean and shaded errors the standard error of the mean. Individual values are represented as single dots. A static right-hand, average or left-hand encoding would be represented by weights of 1, 0.5 and 0, respectively, and are indicated in figure **B**. Different experimental training conditions led to different encoding weights in the two groups (**A**). These weights are developed over the course of the experiment (**B**), especially for the evident group.

shift during exposure toward the respective weighted encoding. This is exactly what we see when analyzing the development of the weights over time in Fig 8B: the ambiguous group had a tendency towards the average encoding throughout the experiment, while the evident group shifted towards the right-hand encoding in the exposure phase and remained there during generalization.

Finally, we analyzed the differences between conditions in the force profiles of the unimanual trials in the generalization phase (Fig 9; Insets in B and G). Whilst there was no group difference for lateral force around peak velocity ($F_{1,14} = 0.631$, $p = 0.440$, $\eta_p^2 = 0.043$), we found both an effect for the condition ($F_{1.432,20.047} = 6.721$, $p = 0.010$, $\eta_p^2 = 0.324$, Greenhouse-Geisser corrected) and an interaction ($F_{1.432,20.047} = 12.275$, $p < 0.001$, $\eta_p^2 = 0.467$, Greenhouse-Geisser corrected). While the ambiguous group scaled the lateral force with increasing velocity in unimanual trials (Fig 9E, $u_f$ vs. $u_s$: $p < 0.001$, $u_m$ vs. $u_s$: $p = 0.012$), the evident group showed similar force profiles for all conditions (Fig 9B, $u_f$ vs. $u_m$: $p = 1.000$, $u_f$ vs. $u_s$: $p = 1.000$, $u_m$ vs. $u_s$: $p = 1.000$). Consequently, the lack of the kinematic information of the right hand led to a generic, partial transfer in the evident group, whereas the ambiguous group showed a partial transfer which matched the trained conditions and underlined the use of the average or weighted encoding to predict the motor output.

## Discussion

Encoding the sensory information of both arms to provide an adequate motor response in our daily lives requires a flexible and adaptive motor memory. Our study focused on the formation of bimanual motor memories and how the experienced states during training affected the encoding of the motor memory and influenced the subsequent motor output. We showed that the human sensorimotor control system is able to flexibly control bimanual actions and form motor memories based on the kinematics of both arms. Two groups of participants adapted to

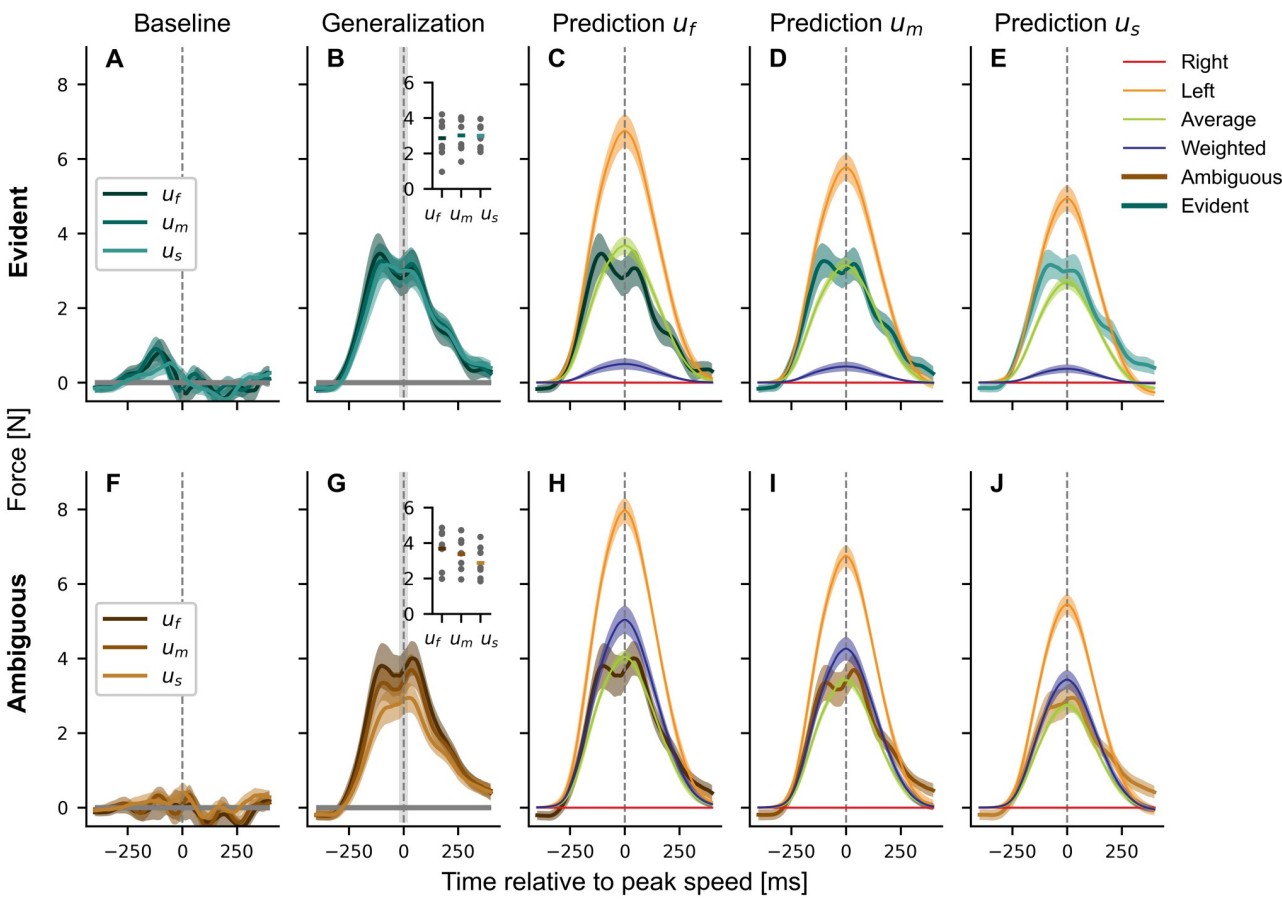

**Fig 9. Force profiles, mean lateral force around peak velocity (dashed grey line) and force predictions for three unimanual conditions $u_f$, $u_m$ and $u_s$.** (**A-E**), The evident group (green). (**F-J**), The ambiguous group (brown). Force profiles are shown as mean values across subjects and blocks for the last baseline and the last generalization block. While the lateral force is around zero in the baseline phase (**A, F**), it is increased after the exposure phase in late generalization (**B, G**). The insets show the mean of the lateral force (horizontal line) and individual data points for each participant and condition over a window around peak velocity (shaded light grey region). While the forces are similar between conditions for the evident group, they are scaled with velocity for the ambiguous group. However, only the ambiguous group showed a force profile similar to weighted encoding (**H** to **J**), while the evident group produced higher forces. Due to the high reliance on right-hand encoding in bimanual trials, the prediction in unimanual trials is low (**C** to **E**). Values represent mean values ± standard error of the mean.

novel dynamics within a bimanual movement task. The evident group trained in conditions in which there was information available that the forces experienced on the left hand depended on the right-hand velocity. The ambiguous group only trained with conditions where this relation was not explicitly presented as both hands made the same movements. The encoding of the motor memory was examined by testing generalization to novel combinations of right and left-hand movement speeds (untrained conditions). Both groups adapted to the novel dynamics during exposure, showing clear adaptation to the trained conditions, but with slightly different patterns of generalization to the untrained conditions. The evident group was able to retrieve the relationship between the right-hand velocity and the force field strength acting on the left hand, as was shown by the shift in encoding weight towards the right hand during exposure, the generalization to novel combinations of the trained conditions (speeds) and the best performance of the weighted encoding model. On the other hand, the ambiguous group mainly learned the force field as a function of both hands with a slight shift towards the left hand (Fig 8), as evidenced by the primarily left-hand encoding on the untrained conditions

during generalization, indicated by the left hand and weighted encoding model (Fig 6). Moreover, the BIC model comparison for all bimanual conditions underlines the selection of the weighted encoding model in the ambiguous group as well (Fig 7). Finally, we tested the transfer of the learned bimanual motor memory to unimanual reaching movements, showing a clear decrease in force for both groups. That is, the transfer to unimanual movements was similar regardless of the shift to either right- or left-hand encoding.

Although the sensorimotor control system performs similar movements over and over again in daily life, none of the movements will be performed identically to the previous movements. Hence, the sensorimotor control system creates a motor memory, adapted to the environmental dynamics, which is able to generalize across similar states. That is, generalization of an existing motor memory helps the sensorimotor control system perform successful actions outside of the specific learned context. The amount and pattern of generalization to a novel context (or different states) differ between dimensions, e.g. between different movement speeds or different movement lengths [18, 20, 25, 29, 33]. As the peak velocity of a movement increases or decreases away from the trained movement, we scale our endpoint force appropriately [18, 29]. There is a strong generalization to shorter movements, but generalization to longer movements is limited unless the occasional interspersed trial at this longer distance is experienced [20]. In our study, we directly trained participants with force fields associated with three movements of different lengths and velocities. Both groups of participants were able to learn to produce the appropriate force compensation for the dynamics with the left hand, scaling to the different movement velocities of the training conditions. While the ambiguous group primarily scaled the forces produced by the left hand according to the left-hand velocity (weighted towards left-hand encoding), the evident group extracted sufficient information to scale (at least partially) the forces in the left hand with the right-hand movement velocities (weighted towards right-hand encoding). Here we have extended previous work [18, 20, 25, 29] by showing that this velocity-dependent scaling of generalization can occur across the two limbs. Importantly, one key aspect of our study is that movement of the non-adapting limb is not only acting as a contextual cue. While associative learning can play a crucial role in motor adaptation [34, 35], the motor memory in the current study is encoded based on the actual kinematics of the movement. This means that participants learned to map the weighted, continuous velocities of the arms to a given force instead of discrete, contextual cues to a perturbation [6, 15, 35, 36].

Generalization also occurs across different movement locations and different movement directions. If the spatial allocation (location) of the movement changes, the amount of generalization found depends on the motor memory representation and the distance away from the trained movement [1, 17, 19, 37]. Similarly, as the angle of the movement changes further away from the trained movement direction we find a decrease in predictive motor output [11, 16, 22–24]. This decrease in the generalization with angular deviation is consistent with the use of local neural basis functions with Gaussian-like tuning as the building blocks of the motor memories [26–29]. The idea is that forming motor memories via these local neural basis functions can provide the flexibility in the tuning of these motor memories to a variety of different states and conditions. Classical, unimanual force field paradigms [1, 14, 22] as well as bimanual force field studies [6, 11, 31] showed that humans are able to directly learn a mapping between the forces perturbing the arm and the velocity of that arm. However, we were able to show that this mapping is dependent on the experienced states during exposure to the force field and that this encoding is flexible enough to be influenced by the training paradigm. The ambiguous group showed a shift towards left-hand encoding which is in line with these classical studies. On the other hand, the evident group used a strong shift to right-hand encoding pattern to generalize to novel combinations of movement speeds of the two hands. This

shows that the sensorimotor control system has sufficient flexibility in its encoding to combine the states of both the right and left arm to encode a new bimanual motor memory. Instead of a rigid distribution between left and right, it employs a highly flexible encoding which reacts rapidly to changing environments. This finding extends previous results of bimanual experiments, which assumed full right-hand encoding [30, 31]. This is also evident in our last step: we removed the kinematic information of the right hand in unimanual trials (where the right hand remains stationary) to see whether any forces were still produced by the left arm. As the evident group was mostly encoding right-hand kinematics in bimanual trials, we expected them to show little transfer. In contrast, we might expect stronger transfer for the ambiguous group as they exhibited a shift towards left-hand encoding. However, transfer from bimanual to unimanual conditions was substantial and similar in both groups. These results underlined previous findings which showed partial transfer when switching from bimanual to unimanual actions and vice versa [6, 11]. Again, this highlighted the flexibility of the human sensorimotor control system while encountering new environments and situations. Interestingly, the partial transfer from bimanual to unimanual control of approximately 50% of the learned peak forces compares well to the reported transfer of 58% by Nozaki et al. [6] and slightly less than the 65% found by Yokoi et al. [11]. Despite the very different training conditions of the two groups in our experiment, which produced a shift towards right-hand encoding for the evident group and left-hand encoding for the ambiguous group, the peak forces were similar for both groups. If this transfer to the unimanual conditions occurs as a byproduct of the final encoding of the learned motor memory, then we would predict two very different levels of transfer in these two conditions, with much higher transfer for the ambiguous group. In addition, we would predict a scaling between unimanual conditions based on the learned scaling from the exposure for the ambiguous group as well. While the latter existed indeed (see Figs 6 and 9) for the ambiguous group and was absent in the evident group, we did not measure a difference between experiments in the total amount of transfer. One possible explanation is that this transfer, or more specifically this tuning of the neural basis functions related to the unimanual motor memory might occur early during the exposure phase when little information about the perturbation is available [38]. This rapid, initial adaptation is then simply maintained in memory throughout the rest of the experiment until probed at the end, and could explain the similarity between both groups. This idea is further underlined by the development of the encoding weights during the experiment (Fig 8B). Initially, the state information for creating the prediction is not evident. During the exposure, participants learned to combine the kinematic inputs from both arms to predict the motor output. Being confronted with a new (unimanual) environment, they might base their predictions on these early states. Despite the overall similar levels of transfer to the unimanual condition, the extensive exposure to the different training conditions produced small but clear differences on the force production in the unimanual trials. Specifically, the ambiguous group, which demonstrated a shift to left-hand encoding for the bimanual trials, showed a small, significant scaling of the lateral forces with the speed of the left hand, matching the scaling during the trained, bimanual conditions. That is, the participants learned that forces were affected by the velocity of the movement, and associated these forces with changes in the velocity of the left hand, as predicted by their shift to left-hand bimanual encoding. In contrast, this scaling was absent for the evident group, where they showed almost identical force profiles for the three different unimanual trials. We propose that this may be due to the stronger reliance on right-hand encoding of the motor memory, and the absence of this additional input that should drive this scaling—the speed or kinematics of the right hand. These differences in the two groups might again reflect our hypothesis that most of the adaptation shown in unimanual trials occurred primarily during early exposure. In addition, force profiles in early exposure show a similar scaling in forces between conditions

for the ambiguous group and no scaling for the evident group (Fig 4B and 4G). We suggest that both groups experienced an initial generic tuning of the unimanual adaptation early in learning, that would not be tuned to the different velocity conditions. Continued exposure to the training conditions would train the right hand encoding in the evident group leaving this generic response intact through the training. In contrast, such exposure in the ambiguous group would train the left-hand encoding in which the force scaled with left-hand velocity, and therefore gradually fine-tune these unimanual responses. Further research measuring unimanual transfer early in exposure is needed to test this hypothesis and understand the processes governing the motor memory formation and the subsequent transfer to unimanual conditions. While the evident group experienced clear differences in the velocities of the hands, allowing them to learn the mapping between right-hand velocity and the resulting forces on the left hand, the ambiguous group was not provided with clear information regarding the force implementation. However, due to noise and variations within the sensorimotor system, there were small differences in velocities between the hands resulting in slightly unmatched movements. This might have allowed them to experience the relationship between right-hand velocity and left-hand forces. These small effects might underlie the small weight of right-hand encoding that was found for the ambiguous group. As this was around 40% across the participants, it is clear that the small trial-by-trial variability in movement velocities was by itself not sufficient for the participants to extract the full force field representation. However, the partial weighting of right-hand encoding even within the ambiguous group supports the findings that higher motor variability and motor noise can be beneficial to adapt to novel environments [39–43].

Recent studies have proposed compound conditioning paradigms from associative learning theory to explain motor adaptation cued by arbitrary stimuli [34, 35]. These studies typically employ a contextual cue that precedes the actual movement [15, 35, 36]. While this is a possible explanation of our findings, we argue that in the current study, the sensorimotor system encoded the weighted velocity of both arms to predict the force instead of explicit cues from the environment. Importantly, one key aspect of the compound paradigm, the additivity principle, states that the combined strength of associated cues is bounded [44–46]. This predicts a negative correlation between associated cues [35]. However, this is not consistent with the current experimental results, specifically the transfer to the unimanual conditions. Given a negative correlation, we would expect higher forces in the ambiguous group, as they would rely more on the left arm movement as a cue, while for the evident group, we would expect lower forces. However, the force levels are similar between groups, rejecting the hypothesis of a bounded associative strength of the contextual cues.

In addition, while we tested generalization on novel combinations of movement speeds, each hand was only tested on conditions in which the training was performed. That is, each hand had previously experienced forces on the fast, medium, and slow movements. Testing generalization of the learned motor memory outside of the trained conditions of both hands (to much faster or much slower movements) would allow us to measure the underlying processes better. This would allow us to determine to what degree the learned motor memory is locally learned compared to full predictive models of the adaptation.

Our findings on the partial transfer of learning between unimanual and bimanual actions, and the difference in motor output between unimanual and bimanual movements, confirming previous results [6, 11], are further underlined by studies reporting altered neural activation patterns in the primary motor cortex between these two actions [7–9]. Recent evidence from a load perturbation study suggested that the activity of neurons in M1 can indeed express two independent representations for unimanual and bimanual movements simultaneously [47]. Further, it has been hypothesized that the cerebellum plays a major role in motor memory

formation [13, 48–51]. The authors state that the cerebellum might host a multitude of different internal models, which could reflect distinct internal models for bimanual and unimanual movements in our experiments. Depending on the context, different models would be retrieved and might control and adjust the subsequent motor output. Apart from the crucial role of M1 and the cerebellum, other areas such as parietal and frontal regions and the spinal cord have been considered to contribute to the sensorimotor integration underlying motor memory formation [52]. This neuro-physiological evidence in combination with the findings from our study underline the need for experiments investigating the interplay between unimanual and bimanual motor memories and their expression on a neuronal level.

To conclude, we were able to show that the human sensorimotor control system is able to encode the sensory information of both hands and flexibly combine the information to control the movement of the other hand. More specifically, the right-hand velocity was used to predict the motor output on the left arm. This highlights that motor memory formation is a highly flexible process, which might occur through the tuning of neural basis functions. The tuning of these basis functions has to take many variables into account, e.g. the kinematics of both arms [11]. This is also underlined by the switch to the unimanual conditions: although the evident group relied mostly on right-hand encoding, they showed substantial transfer—possibly enabled by neural basis functions which have been activated previously during similar states in bimanual movements.

## Materials and methods

### Participants

Sixteen individuals (mean age 25.9 ± 3.0 years, 7 females) participated in this study and were randomly assigned to the evident group (N = 8) or ambiguous group (N = 8). All individuals reported normal or corrected-to-normal vision, no neuropsychological disorders, were free of acute upper limb injuries and right-handed according to the Edinburgh Handedness Questionnaire [53]. Four participants had previous experience performing experiments using a robotic manipulandum.

### Ethical statement

All participants gave their written informed consent and volunteered for the study without any financial compensation. The ethics commission of the Faculty of Medicine at the Technical University of Munich approved the experiments.

### Experimental apparatus and setup

**Apparatus.** We used a bimanual set-up of the two-dimensional, planar robotic manipulandum vBOT [54] to apply state-dependent forces to the arms (Fig 1B). Position and velocity of the handles in the horizontal workspace are recorded via joint position sensors on the motor axis (58SA; Industrial encoders design) and endpoint forces on the left hand were measured by a six-axis force transducer (Nano-25 six-axis force/torque transducers, ATI Industrial Automation, Apex, NC, USA). The sampling rate was set to 1 kHz. The robot and data acquisition was controlled by a customized Microsoft Visual C++ software library under Windows XP [54].

A virtual reality system was used to present visual feedback regarding the task and to prevent the participants from directly viewing their arms. A semi-silvered mirror system, which reflected an attached screen, provided visual feedback of the task in the plane of movement. Visual feedback of each hand's position was indicated by a red cursor with a diameter of 0.5

cm. Start and end targets were represented by yellow circles of 1.0 cm and 1.5 cm diameter, respectively, which turned to white when the cursor was located within their boundaries. Between the target positions, a white cross served as a visual fixation point during the movement. The participants were seated on a height-adjustable chair and fixed by with a four-point safety harness in order to maintain the same upper-body position throughout the experiment. The workspace was located approximately 25 cm in front of the participant's chest and centered with their body. Two air sleds supported each forearm to constrain the arm movement to the horizontal plane and prevent fatigue.

**Course of a trial.** Participants performed bimanual, parallel reaching movements with both arms and unimanual reaching movements with the left arm (right arm stationary; see Fig 1D). The movements of each hand could be short, medium or long lengths. We instructed participants to make straight, natural reaching movements to the targets (or target) while keeping their eye gaze at the fixation cross. At the beginning of each trial, the start circles and end targets appeared on the screen, and the robot moved each hand to the starting circle. Once the two hands were stationary within the start circle for 1000–2000 ms (randomly determined based on a truncated exponential), a beep tone indicated the start of the movements. Participants were asked to make simultaneous movements of both hands to the two targets. Trials ended when both cursors rested in the target position for 600 ms. After each movement feedback was provided about the speed of each hand separately. If participants obtained peak speed within the desired ranges, they were provided with "good" or "great" feedback ("great" occurred when participants' speed was within the middle 50% of the desired range). Desired peak speeds were 75 ± 7.5 cm/s, 60 ± 6 cm/s and 45 ± 4.5 cm/s for long, medium and short reaches respectively, such that the same movement time was required for each reach. If participants did not obtain the desired peak speed, they were told whether they were "too slow" or "too fast". Individuals received a point when both hands met the desired speed and were encouraged to collect as many points as possible. Independent of the score, we included all trials into analysis.

After each trial ended, the robot moved the handles (and participants' hands) back to the next starting positions (start circle) to start a new trial. Movements were self-paced, meaning that participants could rest at any point during the experiment. Every 150 trials a teapot indicated a short break, in which participants were instructed to release the handles and pause for approximately one minute. In the middle of the experiment, we introduced a longer break of approximately 25 min.

We used three different field types in the experiment: null field, force field and error clamp. The force field and the error clamp were only applied to the left hand. The right hand always moved in a null field. In the null field, the robot applied no force on the handle and participants could move freely. In the force field, the manipulandum created a velocity-dependent, counter-clockwise curl force field on the left handle with the following properties:

$$\begin{bmatrix} F_{x_{left}} \\ F_{y_{left}} \end{bmatrix} = \begin{bmatrix} 0 & -13 \\ 13 & 0 \end{bmatrix} \begin{bmatrix} \dot{x}_{right} \\ \dot{y}_{right} \end{bmatrix} \tag{1}$$

such that the forces on the left hand depended on the movement of the right hand. In the error-clamp trials, a mechanical channel [14, 55, 56] was applied. The mechanical channel allows straight movements to the target but restricts lateral movements of the left hand. The

channel trials were defined as:

$$\begin{bmatrix} F_x \\ F_y \end{bmatrix} = \begin{bmatrix} K_x x + D_x \dot{x} \\ 0 \end{bmatrix} \qquad (2)$$

with a stiffness $K$ of 6,000 Nm and a damping $D$ of 20 Ns/m to reduce lateral vibration while moving through the channel. These error clamp trials allowed us to measure the force applied by the participants against the channel wall via the force transducers at the handle.

## Experimental conditions

The experiment consisted of nine bimanual and three unimanual conditions (Fig 1D). In the bimanual (b) conditions, each hand could move one of three different distances [15, 20, 25] cm in the same movement time (700 ms), resulting in a slow (s), medium (m) or fast (f) movement respectively of each hand. All possible combinations were performed, resulting in nine different bimanual conditions. Each condition is named according to the left and right speed requirements, such that a slow movement on the left hand and a fast movement on the right hand would be indicated as $b_{sf}$, whereas a fast movement on the left hand and slow movement on the right hand is indicated as $b_{fs}$.

In the unimanual (u) conditions, the left-hand movement distance was one of 15, 20 or 25 cm, while the right hand remained stationary. Again the desired movement time was fixed to 700 ms for all distances, resulting in a slow (s), medium (m) or fast (f) movement of the left hand. These three conditions were therefore indicated as $u_s$, $u_m$, and $u_f$.

Two groups of participants (evident and ambiguous) performed the experiments and experienced force fields on three of the bimanual conditions. The evident group trained on conditions in which the participants would clearly experience the left-hand forces depending on the right-hand velocity. That is, the evident group, was trained on two conditions in which the left and right hands moved at different speeds ($b_{sf}$, $b_{fs}$) as well as the middle condition ($b_{mm}$). In contrast, the ambiguous group trained on conditions in which this relationship between left-hand forces and right-hand velocity was not clearly apparent. Specifically, the ambiguous group was trained on conditions in which the left and right hands moved at the same speeds ($b_{ss}$, $b_{mm}$, $b_{ff}$). Although in both groups the forces experienced in the left hand depend on the right-hand velocities, the evident group was presented with conditions where the left- and right-hand velocities were clearly different, whereas the ambiguous group would only experience this if the left and right-hand velocities were different according to motor variability.

The experiment itself was divided into three phases (Fig 1A): baseline, exposure and generalization. The baseline phase consisted of six blocks of 48 trials (288 trials total) in the null field. Within the first five baseline blocks, each condition (unimanual and bimanual) was presented three times in the null field (36 null field trials) and once in an error clamp trial (12 error clamp trials). All conditions were randomized across each block. In the last baseline block, participants were only presented with the three training conditions (evident group: $b_{sf}$, $b_{mm}$ and $b_{fs}$; ambiguous group: $b_{ff}$, $b_{mm}$ and $b_{ss}$), with two error clamp trials and 14 null field trials for each of the three conditions (6 error-clamp and 42 null field trials). The exposure phase consisted of 20 blocks of 48 trials (960 trials total) in the force field. These blocks contained the same structure as the final baseline block, with two error clamp trials and fourteen force field trials for each training condition. The final generalization phase consisted of 20 blocks of 48 trials (960 trials total). In the generalization phase, each block consisted of twelve force field trials for each of the three training conditions (36 force field trials) and one error clamp trial for all bimanual and unimanual conditions (12 error clamp trials).

## Data analysis

The data was analyzed with Python 3.8.12, the Sypder IDE (Spyder 5.1.5, The Scientific Python Development Environment) and Jupyter Notebook 6.4.6. Position and force data was filtered with a 5th order, zero lag Butterworth low pass filter with a cut-off frequency of 40 Hz, using SciPy's filtfilt function [57]. To remove any drift of the measured handle forces due to an inconsistent position on the air sled during error clamp trials [58], we subtracted the mean lateral force applied 200 ms to 150 ms before movement onset from the subsequent force vector.

**Kinematic error.** During null field and force field trials, we used the maximum perpendicular error (MPE) as a measure of adaptation to the force field. It reflected the signed maximum perpendicular distance between the handle position and the straight line connecting the center of the start and end targets. For plotting purposes, we averaged the maximum perpendicular error over a bin of 12 trials.

**Force compensation.** The force compensation (FC) is a scalar which depicts the amount of adaptation to the force field, by measuring the imposed forces acting laterally on the handle on the clamp trials [56]. The force compensation value is calculated as the regression coefficient of a linear regression between the actual force output and the predicted force profile required to perfectly compensate the perturbation [56]. In our study, the force field applied to the left hand depended on the velocity of the right handle. Therefore, the predicted force profile for bimanual conditions is calculated as the product of the right-hand velocity and the force field strength (13 Ns/m). Importantly, a force field was never applied during any unimanual conditions.

**Force profiles and peak forces.** In order to evaluate the shape and timing of feedforward, predictive forces during channel trials, we aligned the force profile of the left hand to the peak velocity of the right hand and clipped the time window to 400 ms before and after peak velocity [58]. Peak forces were assessed by calculating the average of the left force profile in a window of 20ms around peak velocity of the right arm in bimanual trials and around peak velocity of the left arm in unimanual trials.

**Predicted outcomes and encoding weights.** In our experiment, the force field was applied to the left hand according to the velocity of the right hand. Here we examined whether adaptation resulted from learning this relationship (dependence on the velocity of the right hand), simply independent learning of the force field within a single limb (ignoring the relationship with the right-hand velocity) or different combinations of both arms. These combinations could entail average encoding by weighting each input equally or a weighted encoding by a variable weight between arms. First, we fitted the different predictions to the generated force profiles during generalization in the trained conditions. We fitted the right, left, average, and weighted encoding models to the force profiles by applying a nonlinear regression model (minimize from SciPy) to our force and velocity data defined by:

$$\mathbf{F} = \alpha_{right} \times \dot{\mathbf{x}}_{right} \times B \tag{3}$$

$$\mathbf{F} = \alpha_{left} \times \dot{\mathbf{x}}_{left} \times B \tag{4}$$

$$\mathbf{F} = \alpha_{average} \times ((\dot{\mathbf{x}}_{left} + \dot{\mathbf{x}}_{right}) \times 0.5 \times B) \tag{5}$$

$$\mathbf{F} = \alpha_{weighted} \times (\omega \times \dot{\mathbf{x}}_{right} \times B + (1 - \omega) \times \dot{\mathbf{x}}_{left} \times B) \tag{6}$$

where $\dot{\mathbf{x}}_{right}$ and $\dot{\mathbf{x}}_{left}$ were the velocity vectors of the right and left hand, $B$ was the force field strength with 0.13 N and $\mathbf{F}$ was the actual force profile. The $\alpha$ values represented the slope of

the regression between the different encoding patterns of velocity and the generated force. Initial $\omega$ in (6) was set to 0.5, which corresponded to an equal distribution between left and right prediction. The weights were unrestricted. A weight of 1.0 would reflect perfect right-hand encoding whereas a weight of 0 would reflect perfect left-hand encoding. Consequently, we calculated the average $\omega$ value for each participant and the results are depicted in Fig 8A.

To evaluate the generalization and transfer to unimanual conditions, we used the $\alpha$ and $\omega$ values retrieved from the fit on trained conditions to generate the predictions for each generalized condition by multiplying the generated velocities with these weights. The predicted force profiles and the predicted force compensation are depicted in Figs 6 and 9, the predicted peak forces in S1 Fig Appendix. To quantify the difference between generated force compensation and forces profiles, we calculated the mean squared error for each subject and condition (see Fig 6 and S1 Fig).

Lastly, we calculated the development of the weighted encoding for each group by fitting Eq 6 to smaller blocks of 96 trials through the exposure and generalization phases.

## Model comparison

To compare between the models, we conducted two BIC calculations. The first BIC was calculated for the trained conditions in trials of the generalization phase to avoid any effects of initial learning. We compared the right hand, left hand, average (all one parameter), and weighted encoding model (two parameters) to a non-parametric model with a static $\alpha = 0.822$, which was the average $\alpha$ across models and participants. We calculated a second BIC across all bimanual conditions during generalization to evaluate if the models can explain generalization as well. The results are depicted as positive BIC improvements compared to the non-parametric, standard model. In addition, we calculated $R^2$ values for each model.

## Statistical analysis

We used JASP (version 0.16) [59] for our statistical analysis. To compare velocities between arms within an experiment, we calculated separate repeated measures ANOVA for each group with the factor condition. To test for adaptation, we conducted a mixed ANOVA on the dependent variables force compensation and maximum perpendicular error with the within-subjects factor phase (3 levels) and the between-subjects factor group (2 levels). The three levels for force compensation were late baseline, late exposure and late generalization, and for the maximum perpendicular error these were early exposure, late exposure and late generalization. In addition, we calculated a 2-factor ANOVA with repeated measures for the dependent variable lateral peak force. After fitting the four models to the data from trained conditions, we compared the $\alpha$ values with a repeated measures ANOVA (within-subject factor model and between-subject factor group). For the comparison of $\omega$ between groups, we used an independent t-test. To assess generalization, we calculated two repeated measures ANOVAs for the generated force compensation with within-subject factor condition and between-subject factor group, as well as the mean squared error with model and group for within-subject and between-subject factor, respectively. Finally, one repeated measures ANOVA with group as between-subjects factor and unimanual condition as the repeated measures factor was conducted to test for differences in lateral force around peak velocity in unimanual trials. A Mauchly test [60] and Levene's Test for Equality of variance [61] were applied to check for sphericity and homogeneity, respectively. In the case that the Mauchly test was significant, the degrees of freedom were adjusted using a Greenhouse-Geisser correction. We used Bonferroni post-hoc tests to determine any differences between the levels. We reported $\eta_p^2$ as the effect size. For t-tests, we tested normal distribution via the Shapiro-Wilk test and presented the

effect size as Cohen's d. If the equal variance assumption was violated, we conducted a Mann-Whitney U-test alternatively. The $\alpha$-level was set to .05 in all tests and results were reported as means and standard errors of the mean.

## Supporting information

**S1 Fig. Peak force predictions and generated peak fores in bimanual conditions.** The horizontal lines depict the group average and dots individual participants. The right-hand encoding shows the required force induced by the force field respective the individual adaptation for each participant. (**A**), The ambiguous group generated peak forces, which deviated from the required forces (red), especially in the conditions $b_{mf}$, $b_{sf}$, and $b_{sm}$. (**B**), The evident group scaled the peak forces inversely during the trained conditions, and were able to generalize to the generalized conditions (as indicated by their adherence to the right-hand encoding). However, generalization was less strong for $b_{ff}$ and $b_{fm}$.
(TIF)

**S2 Fig. Mean squared error (MSE) for different models for force profiles in generalized conditions.** The results show individual data (dots) and the group average (horizontal line) of the MSE during the generalization phase. (**A**), The evident group showed the lowest MSE for right-hand, average and weighted encoding, with being significantly lower compared to the left-hand encoding. (**B**), In contrast, the ambiguous group had a lower MSE for left-hand, average and weighted encoding.
(TIF)

## Acknowledgments

We thank Marion Forano and Filipa Pereira for initial work on an early version of this study, Sae Franklin for assistance with running the experiments, and Clara Günter, Raz Leib and Sae Franklin for their helpful feedback on the research. We also thank all the volunteers who participated in the study and contributed with their patience to the long experiments.

## Author Contributions

**Conceptualization:** Jonathan Orschiedt, David W. Franklin.

**Data curation:** Jonathan Orschiedt.

**Formal analysis:** Jonathan Orschiedt.

**Investigation:** Jonathan Orschiedt, David W. Franklin.

**Methodology:** Jonathan Orschiedt, David W. Franklin.

**Resources:** David W. Franklin.

**Software:** Jonathan Orschiedt, David W. Franklin.

**Supervision:** David W. Franklin.

**Visualization:** Jonathan Orschiedt.

**Writing – original draft:** Jonathan Orschiedt, David W. Franklin.

**Writing – review & editing:** Jonathan Orschiedt, David W. Franklin.

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
