## [Decision Letter · Decision Letter 0]

12 Jul 2023

Dear Prof. Franklin,

Thank you very much for submitting your manuscript "Learning context shapes bimanual control strategy and generalization of novel dynamics" for consideration at PLOS Computational Biology.

As with all papers reviewed by the journal, your manuscript was reviewed by members of the editorial board and by several independent reviewers. The reviewers found that the study was well designed and well conducted, with mostly clear results and conclusions, but did raise some substantial concerns. In particular, the application of computational models is not as thorough as would normally be expected for papers published in PLOS Computational Biology. Given that the application of computational models is central to this journal we have to regard this as a significant weakness of the paper. We would, however, be willing to consider a revised version of the paper if you can address this, along with other weaknesses highlighted by the reviewers.

One suggestion for streamlining the computational part of the paper is that presenting and fitting a multitude of closely related models (Equations 3-7) is overly complicated when in fact all the models considered are special cases of a general model in which the predicted force is a linear combination of left and right hand velocities. It may be more straightforward to fit this two-parameter model (which is basically equivalent to the weight model) and interpret the estimated parameters, rather than comparing 5 distinct models, some of which are quite arbitrarily chosen (e.g. the average and subtract models). A clearer evaluation and presentation of how well this model accounts for behavior across the different generalization conditions (ideally on conditions outside those used to fit the model) would also help improve the modeling results.

I wanted to add a few other comments on top of those of the reviewers. First, before Jackson and Miall, there was also a paper from Bays and Wolpert (J Neurosci, 2006) which considered a similar paradigm and ought to be cited. You might also look at Krakauer et al., PLOS Biology, which examined generalization of shoulder/wrist movements, rather than left hand/ right hand, but contains a similar idea. Third, the attribution of the force to either the left or right hand is very reminiscent of compound conditioning paradigms in which e.g. a light and a tone are presented together, then one is subsequently removed. See e.g. Gershman 2015, PLOS Computational Biology for a review of models of this type of task, which might be pertinent. A model along these lines was also included in Krakauer et al., 2006, and somewhat related ideas are discussed in Avraham eLife, 2022. You may find that these are interesting parallels to at least discuss.

In light of the reviews (below this email), we would like to invite the resubmission of a significantly-revised version that takes into account the reviewers' comments. In particular, in this revision, we would expect the computational parts of the paper to be substantially improved, although all other reviewer concerns should also be addressed. We cannot make any decision about publication until we have seen the revised manuscript and your response to the reviewers' comments. Your revised manuscript is also likely to be sent to reviewers for further evaluation.

Also note that, alternatively, if you feel that your results can stand alone without relying strongly on computational modeling approaches (which may well be the case), then you might prefer to submit your work to a different journal that places less emphasis on using computational methods.

When you resubmit, please upload the following:

Sincerely,

Adrian M Haith

Academic Editor

PLOS Computational Biology

Marieke van Vugt

Section Editor

PLOS Computational Biology

Reviewer's Responses to Questions

**Comments to the Authors:**

Reviewer #1: Summary:

The authors presented clear evidence that during bimanual movement, one arm can refer to

the other arm's motion state when learning to resist the mechanical load induced by the

other arm. The authors also showed that this attribution to the arms’ motion states is

dependent on the degree how the motion states of two arms are correlated; If the motion

states of the arms are de-correlated, one can learn the correct association between the load

to the left arm and motion of the right arm. On the other hand, when the states are perfectly

correlated (i.e., both arms were moving in the same way), the load induced by the right arm

motion was partially attributed to the motion of the left arm. Although this finding is

somewhat expected, a more interesting finding is that despite such a difference in attribution,

the overall transfer of bimanual learning to the unimanual movement was less affected by

the training context. The results presented are informative for the motor learning community.

Overall, the manuscript is well-written, and the results are clear. The authors did a

good job addressing the concerns raised in the previous round. I only have several minor

points that can be easily addressed.

Minor Comments:

Line#157-158 and line# 266-267: Given the results of unimanual conditions (Fig. 7E), it looks

ambiguous group shows modulation by the speed in the early exposure phase (Fig. 4G).

Though this may be simply due to the use of different ANOVA, a brief mention of this could

be helpful.

Line# 176-260: ANOVA results are missing. Referring to appropriate figure panels (e.g., Fig.

5A, B) would largely help readers to understand the results of bimanual conditions.

Especially, Figures 5B and 5D seem not referred to in the Results section.

Line#192-193: B is omitted here.

Figure 5C and D: If I correctly understood, the models were fitted to training condition data

and then tested on the data for generalization conditions. It is unclear whether the MSEs

presented here include that for the training data. In the context of cross-validation, the MSEs

for the fitting (i.e., MSEs for training condition data) should be excluded from the evaluation.

Figure 5D: Missing x-labels

The presentation of Figure 5 could be improved. For instance, the data for bimanual and

unimanual conditions could go into separate figures, as the current version is too visually

crowded. Also, using lines in Fig. 5C and D is not very informative. Simply using dots

arranged horizontally can improve visibility in the same space.

Panel D can be only for unimanual conditions and probably go to separate figures with

unimanual peak force data (from panels A and B).

As there are separated panels for the two groups, actual force values do not need different

colors. Currently, it is visually hard to detect lines in similar colors (e.g., red/orange lines over

brown shade). Gray shade could be nice for getting good contrast with all the colors used.

Line#609: Referring to wrong Eq. # (should be seven instead of 5).

Line#594-619 (regression analysis): Were the alpha values also constrained to [0,1]? If so, it

might be unfair to the subtraction coding model (perhaps the presence of B requires the

constraint on alpha, but the B in the models looks redundant). Please clarify.

Please also specify how unimanual conditions were simulated (by simply setting x_dot_right

to 0?).

Reviewer #2: In this work, the authors probe the influence of speed or movement length as a factor in adopting a bimanual control strategy. They perform an experiment wherein the speed of right hand movement determined the lateral force applied to the left hand. Two experimental groups were employed, both utilizing the velocity of right hand movement to determine the left-hand forces while the right hand always moved in a null field. In the “ambiguous” group, both hands moved at identical speeds and distances, creating ambiguity for the sensorimotor system in utilizing the sensory state of either hand to plan subsequent movements. In contrast, for the “evident” group, there was a distinction in speed between the hands, making it evident to the sensorimotor system that right hand speed was more relevant for predicting the force-field applied to the left hand. Generalization trials, incorporating interspersed error clamp catch trials, were used to assess the encoding scheme utilized in forming the bimanual memory. The authors report that the evident group employed a right hand encoding scheme, whereas the ambiguous group adopted an average or weighted average encoding scheme for bimanual memory formation.

1. How the motor system adapts to perturbations applied during bimanual movements is not well understood, and the literature on this topic is also limited. The study therefore addresses an important issue, but the findings are hardly profound or ground-breaking. The larger conclusion that training conditions strongly influence the encoding of a motor memory, and hence also shape subsequent generalization, is well-known from work in unimanual movements. The current study extends this to bimanual action conditions, but does not break new ground in terms of deepening our understanding about the influence of training conditions.

2. I also think that the authors have over-interpreted their results in favour of what has been ultimately reported. Specifically, it is not at all clear to me why the average encoding isn’t the best predictor of how people adapt left arm dynamics in either group. The authors do consider this possibility for the ambiguous group (and this makes sense), but I am not convinced that the right hand encoding does better than the average encoding for the evident group. All the plots of figure 5 and the statistics also point to average encoding doing better even for the evident group. The authors are requested to present key metrics that will convince a reader that

3. Control groups with force fields applied to the left arm under unimanual conditions are sorely missing from the study. While it may be obvious that predictive force development will be associated only with left arm kinematics in this case, it is important to know how the left arm contribution during bimanual actions, particularly in the ambiguous case, emerges/changes relative to the unimanual condition.

4. The modelling Results are very under-reported. The authors have made somewhat sweeping statements about one encoding model being better or worse than another. However, no statistical comparison across different models is available. The authors must provide a solid comparison across models (using AIC, etc.) so that one can make more informed and reliable conclusions about which encoding method works best in the two cases.

5. This study has a very limited sample size – only 8 subjects per group. Our field of work has gained some disrepute due to small sample sizes and most labs are striving to do better. The authors are therefore requested to provide a justification for why such a small sample size is good enough for making sound conclusions.

Minor:

1. Is the right arm encoding line missing from the bff condition in Figure 5B? I cant seem to find the red line.

2. line 609: "Initial ω in (5) was set to 0.5". There is no ω in eq. 5.

**Have the authors made all data and (if applicable) computational code underlying the findings in their manuscript fully available?**

Reviewer #1: **No: **The authors state the data will be made available on Figshare before publication.

Reviewer #2: **No: **I could not find a statement on data and code availability.

PLOS authors have the option to publish the peer review history of their article (what does this mean?). If published, this will include your full peer review and any attached files.

Reviewer #1: No

Reviewer #2: No
---

## [Decision Letter · Decision Letter 1]

15 Nov 2023

Dear Prof. Franklin,

We are pleased to inform you that your manuscript 'Learning context shapes bimanual control strategy and generalization of novel dynamics' has been provisionally accepted for publication in PLOS Computational Biology.

Best regards,

Adrian M Haith

Academic Editor

PLOS Computational Biology

Marieke van Vugt

Section Editor

PLOS Computational Biology

Reviewer's Responses to Questions

**Comments to the Authors:**

Reviewer #1: The authors have nicely addressed my previous comments. I have no further comment to add.

Reviewer #2: I think the authors have a done a good job addressing the concerns raised. I would recommend accepting the manuscript after a few typos in the document are fixed.

**Have the authors made all data and (if applicable) computational code underlying the findings in their manuscript fully available?**

Reviewer #1: Yes

Reviewer #2: Yes

PLOS authors have the option to publish the peer review history of their article (what does this mean?). If published, this will include your full peer review and any attached files.

Reviewer #1: No

Reviewer #2: No

---

## [Editor Report · Acceptance letter]

4 Dec 2023

PCOMPBIOL-D-23-00784R1 

Learning context shapes bimanual control strategy and generalization of novel dynamics

Dear Dr Franklin,

I am pleased to inform you that your manuscript has been formally accepted for publication in PLOS Computational Biology. Your manuscript is now with our production department and you will be notified of the publication date in due course.

With kind regards,

Lilla Horvath
